# Factors associated with pneumococcal nasopharyngeal carriage: A systematic review

**Eleanor Frances Georgina Neal** [1,2]*, **Jocelyn Chan** [1,2], **Cattram Duong Nguyen** [1,2], **Fiona Mary Russell** [1,2]

**1** Infection & Immunity, Murdoch Children's Research Institute, Royal Children's Hospital, Parkville, VIC, Australia, **2** Department of Paediatrics, The University of Melbourne, Parkville, Australia

* eleanor.neal@mcri.edu.au

**Data Availability Statement:** All data and related metadata underlying the findings reported are included in the submitted article and its supplementary information files.

## Abstract

Pneumococcal disease is a major contributor to global childhood morbidity and mortality and is more common in low- and middle-income countries (LMICs) than in high-income countries. Pneumococcal carriage is a prerequisite for pneumococcal disease. Pneumococcal conjugate vaccine reduces vaccine-type carriage and disease. However, pneumococcal carriage and disease persist, and it is important to identify other potentially modifiable factors associated with pneumococcal carriage and determine if risk factors differ between low, middle, and high-income countries. This information may help inform pneumococcal disease prevention programs. This systematic literature review describes factors associated with pneumococcal carriage stratified by country income status and summarises pneumococcal carriage rates for included studies. We undertook a systematic search of English-language pneumococcal nasopharyngeal carriage studies up to 30th June 2021. Peer-reviewed studies reporting factors associated with overall pneumococcal nasopharyngeal carriage in healthy, community-based study populations were eligible for inclusion. Two researchers independently reviewed studies to determine eligibility. Results are presented as narrative summaries. This review is registered with PROSPERO, CRD42020186914. Eighty-two studies were included, and 46 (56%) were conducted in LMICs. There was heterogeneity in the factors assessed in each study. Factors positively associated with pneumococcal carriage in all income classification were young age, ethnicity, symptoms of respiratory tract infection, childcare attendance, living with young children, poverty, exposure to smoke, season, and co-colonisation with other pathogens. Breastfeeding and antibiotic use were protective against carriage in all income classifications. Median (interquartile range) pneumococcal carriage rates differed by income classification, ranging from 51% (19.3–70.2%), 38.5% (19.3–51.6%), 31.5% (19.0–51.0%), 28.5% (16.8–35.4%), ($P$ = 0.005) in low-, lower-middle, upper-middle, and high-income classifications, respectively. Our findings suggest that where measured, factors associated with pneumococcal nasopharyngeal carriage are similar across income classifications, despite the highest pneumococcal carriage rates being in low-income classifications. Reducing viral transmission through vaccination and public health interventions to address social determinants of health would play an important role.

**Funding:** The authors received no specific funding for this work.

**Competing interests:** The authors have declared that no competing interests exist.

## Introduction

*Streptococcus pneumoniae* is a leading cause of invasive and non-invasive diseases, including meningitis, sepsis, pneumonia, and acute otitis media [1–4]. Most cases occur in children under five and the elderly [4, 5]. The epidemiology of pneumococcal disease varies by region, with the highest incidence and mortality rates across Southeast Asia and Africa [4]. Pneumococcal colonisation precedes pneumococcal disease [6–8]. As with pneumococcal disease, pneumococcal carriage epidemiology varies by age. Carriage rates tend to be highest amongst children in the second year of life [9, 10].

Understanding the factors associated with pneumococcal carriage is essential to developing appropriate public health intervention strategies to prevent pneumococcal disease and evaluating the impact of the pneumococcal conjugate vaccine (PCV), particularly among subpopulations most at risk [11]. Previous reviews of pneumococcal carriage prevalence, serotype distribution, and impact of PCV on carriage, including data from children and adults in low- and lower-middle-income countries and sub-Saharan Africa [12, 13], reported a variety of risk factors for pneumococcal carriage, including seasonality [14, 15], rural residence, age, crowding, co-colonisation with respiratory pathogens, presence of respiratory tract infections, and comorbidities such as human immunodeficiency virus (HIV) [12, 13]. A previous systematic review reported that pneumococcal carriage rates were generally higher in low-income countries than middle-income countries; however, it is unknown if risk factors for carriage differ across low- middle-and high-income countries [12].

PCV is the most effective measure to prevent pneumococcal disease and is an intervention known to reduce carriage of vaccine serotypes [16–18]. In some settings, such as Malawi and Papua New Guinea, vaccine-serotypes have continued to circulate post-PCV introduction [19, 20]. Additionally, replacement with non-vaccine serotypes in carriage, and to a lesser extent disease, has occurred in some settings post-PCV introduction [21–23]. In addition to introducing PCV, other interventions can also reduce pneumococcal disease [24–26]. This study aims to identify risk factors for overall pneumococcal carriage to determine if there are other potentially modifiable exposures, in addition to PCV vaccination. No systematic literature review on the risk factors of pneumococcal carriage has been published previously. Therefore, we undertook a systematic literature review to describe pneumococcal nasopharyngeal carriage risk factors in community-based populations, stratified by country income classification, and summarised the reported pneumococcal carriage rates of included studies.

## Methods

### Protocol and registration

A systematic literature search was undertaken and reported according to the PRISMA guidelines [27]. The review protocol was registered with the International Prospective Register of Systematic Reviews, registration CRD42020186914.

### Eligibility criteria

Peer-reviewed studies reporting factors associated with pneumococcal nasopharyngeal carriage, published in English before July 2021, were eligible for inclusion. As risk factors assessed in primary studies were unlikely to be similar across all pneumococcal carriage studies, we limited our review to studies of healthy, community-based populations in an attempt to select studies with as similar as possible exposures. For transparency and completeness, we also present the results for all factors assessed for association with pneumococcal carriage by each included primary study. We included community-based studies in which nasopharyngeal

swabs had been taken and processed to detect pneumococci and in which statistical analyses had been conducted to determine factors associated with overall pneumococcal carriage.

We excluded animal studies, letters, case reports, editorials, comments, diagnostic articles, and intervention studies where the intervention was likely to impact the rates of pneumococcal carriage. We also excluded studies if the description of the study design and laboratory methods was inadequate (lacked inclusion/exclusion criteria, lacked an adequate description of settings, populations, or laboratory methods); if risk factors were assessed via univariable analysis only; if a *P*-value only was reported; if the sample size was less than 30 participants; if the total study population was non-community (e.g., hospital, prison, or childcare) or had underlying comorbidities (e.g., HIV); and if factors associated with VT or non-vaccine-type (NVT) pneumococcal carriage only were reported.

## Information sources and search

We searched Medline (Ovid) and Embase (Ovid) on 30[th] June 2021, using thesaurus and keywords. We adapted the Medline search strategy for use in the Cochrane library. PubMed was searched using a combination of search strings for *S. pneumoniae*, carriage, colonisation, and nasopharynx. The search strategies are available in S1 Text.

## Study selection

In the first instance, two reviewers (EFGN and JC) independently screened titles and abstracts of retrieved articles. We excluded duplicates and articles unrelated to the primary subject matter. Secondly, we assessed the eligibility of selected full texts. Reviewers resolved discordant eligibility results by discussion.

## Data collection

Study data were collected and managed using REDCap electronic data capture tools hosted at the Murdoch Children's Research Institute or entered directly into pre-prepared tables [28]. EFGN exported quality assessment and extracted study data in REDCap to Stata 16.1 for cleaning and analysis [29].

Extracted data included: first author; year of publication, research aim, study design; study year(s); country; setting (e.g., semi-urban); swab material; pneumococcal identification method; age group; sample size; citation of 2003 or 2013 World Health Organization (WHO) recommendations for detecting pneumococci in upper respiratory tracts [30, 31]; pneumococcal carriage rates; factors associated with pneumococcal carriage, with associated effect size estimates, and where reported, 95% CI and *P*-value; analysis method to determine associations; and variable selection method for models used to assess factors associated with pneumococcal carriage (e.g., empirical based on *P-value* thresholds). If the study design was not stated, we compared the described methods and classified them according to standard study design definitions [32]. For each study, we used World Bank data to determine the income status (low, lower-middle, upper-middle, and high) of the study country during the study period [33]. We categorised each study by WHO region (Africa, Americas, Eastern Mediterranean, European, South-East Asia, and Western Pacific) in which the study was conducted [34]. Reviewers contacted the authors of some eligible primary studies to obtain additional information. We used a Kruskal-Wallis test to compare overall pneumococcal nasopharyngeal carriage rates by income classification (low, lower-middle, upper-middle, and high), as this is a rank-based, non-parametric method for testing differences between two or more categorical, independent groups on a continuous (or ordinal) outcome (where study-level carriage rates were treated as continuous). We present a narrative summary of the results.

## Quality of studies

The quality of each study was assessed using the relevant NIH Study Quality Assessment Tool [35]. We modified the tool by replacing the question "Were outcomes assessed using valid and reliable measures, implemented consistently across all study participants?" with an assessment of whether or not individual studies complied with the 2003 or 2013 World Health Organization standard method for detecting upper respiratory carriage of pneumococci, noting some studies were conducted or designed before the publication of these guidelines [30, 31]. One reviewer (EFGN) independently assessed quality and bias and extracted data. A second reviewer (JC) independently assessed quality and bias and extracted data from a random subset (20%) of articles for data validation. Reviewers resolved discordance through discussion.

# Results

## Search results

We identified 2,558 articles and excluded 1,015 as duplicates. The remaining 1,543 underwent title and abstract screening (Fig 1). Of those, 1,083 were excluded. We assessed the full text of 460 articles, retaining 82 for inclusion in this systematic review.

## Characteristics of included studies

Most (81/82, 99%) studies were observational. Sixty-six (80%) were cross-sectional studies [14, 15, 36–98] (S1 Table). There were six (7%) cohort [99–104], four (5%) nested-cohort [105–108], two (2%) longitudinal [109, 110], and two (2%) nested-longitudinal [111, 112] study designs. There was one nested case-control [113] and one nested cross-sectional study [114]. One study reported secondary analysis using data from a randomised controlled trial [115].

All income settings and WHO regions were represented (S1 Table). The majority (36/82, 44%) were conducted in high-income countries across the Americas [66, 71–80, 84, 107], Europe [81, 85–92, 102, 108, 115], and the Western Pacific [93–95, 97, 98, 103, 104, 110]. One study was conducted across two income classifications and WHO regions: high-income Israel in the WHO European region and lower-middle-income West Bank and Gaza in the WHO Eastern Mediterranean region [96].

Study design, methods, and participant characteristics varied across studies (S1 and S2 Tables). The age groups of participants ranged from neonates to adults, however, the majority (60/82, 73%) focused on infants and toddlers [41, 47–49, 56, 60, 81, 94, 98, 99, 108, 113, 115] or children, with varying ages [14, 36, 38, 44, 45, 51, 52, 57, 59, 63–66, 70–76, 78–80, 83–87, 89, 90, 92, 93, 96, 100, 103, 104, 107, 110]. Study authors measured a variety of participant characteristics. While some, such as sex, age, and recent history of respiratory tract illness, were measured in studies from all income classifications, no single characteristic was measured in all studies. Differences in characteristics measured may reflect the cultural or income context of studies and participants' ages. For example, childcare attendance was measured in approximately 81% (29/36) of studies conducted in high-income countries, but only 18% (3/17), 9% (1/11), and 38% (7/18) of studies conducted in low, lower-middle, and upper-middle-income countries.

The type of statistical model used to investigate the factors associated with pneumococcal carriage was similar across studies, most commonly (72/82 studies, 88%) logistic regression (S1 Table). The methods used for variable selection varied widely, including empirical only, *a priori* only, or a combination of empirical and *a priori* (S1 Table). Empirical variable selection (alone or in combination with *a priori* selection) was driven by *P*-values, with thresholds for inclusion ranging from <0.1 to <0.5.

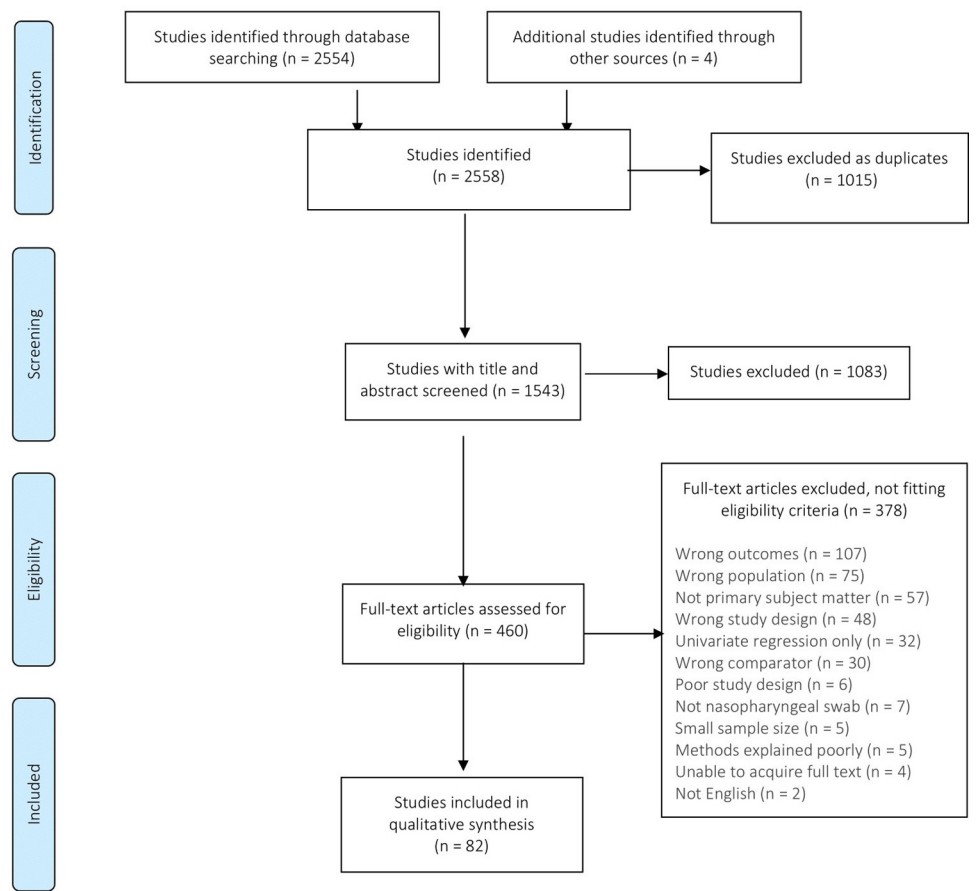

**Fig 1. PRISMA flow diagram for study selection.**

Various methods were used to detect *S. pneumoniae* (S3 Table). Thirty-seven (45%) studies referred to either the 2003 [14, 15, 30, 36, 39, 40, 43, 49, 53–55, 60, 66, 82, 85, 86, 88, 92, 94, 95, 99, 105, 108, 111, 115] or 2013 [31, 37, 38, 44, 46–48, 50, 67–69, 106, 109, 114] WHO methods to detect pneumococci.

## Quality assessment

The quality assessment results are shown in S4 Table [35]. Based on the overall assessment, approximately two-thirds of the studies were considered fair (50/82, 59%) and over a third good quality (30/82, 37%) quality. Two included studies were considered poor quality. Underlying reasons for poor quality related to a lack of sufficient detail in study methods. A cross-sectional study conducted in Hong Kong was considered poor quality as the methods were unclear, including a lack of clarity around the inclusion criteria and insufficient descriptions of pneumococcal carriage detection and statistical methods [110]. In this study, it was unclear how multivariable logistic regression models were built, there was no discussion regarding variable selection, and the only indication that a multivariable model had been used was in the abstract [110]. A cross-sectional study from Bolivia was considered poor quality, as it lacked a clear research question, the inclusion and exclusion criteria were unclear, no sample size calculation was included, and the description of statistical methods was insufficient [54]. Studies were not excluded based on poor quality to ensure transparency and completeness of reporting from all studies identified as relevant to the review [116].

### Results of individual studies

S5 Table shows the factors assessed for associations with pneumococcal carriage, stratified by World Bank income classification [33]. Reported risk factors for pneumococcal carriage were demographic (e.g., young age, symptoms of respiratory tract infection, ethnicity, sex), environmental or household-related factors (e.g., living with siblings or other young children, exposure to tobacco and cooking smoke, attendance at childcare, residential location, season).

**Factors assessed for association with pneumococcal carriage.** Factors assessed for association with pneumococcal nasopharyngeal carriage varied by study (S5 Table). Some factors were measured and assessed in all income classifications, including age [36, 38, 40–43, 45, 46, 50, 52–62, 64–68, 70–78, 80–84, 86, 89, 90, 92–99, 101, 104, 107, 109–111, 115], antibiotic use [14, 36, 37, 40, 41, 45–47, 52, 60, 63, 66, 70, 73–79, 81, 82, 84, 92, 93, 97, 101, 104, 107, 109, 115], breastfeeding (including current, exclusive, history, and duration of) [39, 41, 49, 58, 73, 75, 83, 90, 97, 103, 104, 107, 108, 112, 115], childcare attendance [36, 41, 42, 52, 56, 57, 63, 64, 66, 70, 72–76, 78–81, 83–90, 92, 93, 96–99, 101, 103, 104, 108, 115], exposure to cigarette and cooking smoke [39–42, 46–49, 51, 53, 55, 60–62, 67, 82, 83, 86, 88, 91, 92, 97, 99, 106–108, 110, 112], ethnicity [39, 41, 47, 49, 67, 68, 78, 83, 88, 92, 100, 110], family size, and household numbers [40, 45, 52, 63, 68, 75, 96], crowding or proxies for crowding [39, 45, 55, 65, 77, 88, 100, 110], residential location [45, 47–50, 54, 57, 60, 61, 67–69, 77, 82, 88, 91, 96], respiratory tract infection symptoms [14, 15, 37, 40, 44–47, 49, 53, 56–58, 64, 67, 68, 73–75, 78, 79, 82, 84, 91, 92, 94, 95, 97, 100, 101, 104, 109, 115], participant sex [36–38, 40, 41, 45–47, 53–55, 58, 61, 62, 67, 68, 82, 83, 88–90, 92, 95, 98, 102, 107, 112, 115], and living with siblings or other young children [41, 42, 44–48, 55, 56, 63, 66, 67, 69–71, 73, 74, 78, 79, 81, 83, 84, 86–89, 92–94, 97, 98, 103, 104, 107, 108, 110, 112, 115]. Some risk factors were assessed more frequently than others, depending on income classification. For example, some form of childcare attendance was assessed for association with pneumococcal carriage in 81% (29/36) [66, 72–76, 78–81, 83–90, 92, 93, 96–98, 103, 104, 108, 115], 39% (7/18) [56, 57, 63, 64, 70, 99, 101], 9% (1/11) [52], and 18% (3/17) [36, 41, 42] of studies from high, upper-middle, lower-middle, and low-income countries, respectively. No single risk factor was assessed for association with carriage in all studies, and some risk factors, such as prematurity [41], malnutrition or body mass index [39, 44, 46, 62], and infant mode of delivery [47, 69] were assessed in one or two income classifications only.

**Factors associated with pneumococcal carriage in low-income countries.** In low-income countries, demographic factors commonly associated with pneumococcal carriage included young age [36, 41, 43, 45, 111, 113], and ill-health around the time of the survey, including symptoms of respiratory infections [14, 40, 45], or other unspecified illnesses [38]. Any treatment for illness and antibiotic use in the two or three weeks preceding the survey was protective against pneumococcal carriage in three studies [14, 15, 113], while five studies with broader definitions of antibiotic use reported no association with pneumococcal carriage [36, 37, 40, 41, 45].

One study reported co-colonisation with non-capsulate *Haemophilus influenzae* [15] and malnutrition [44] to be positively associated with carriage–no other study from low-income countries assessed these potential risk factors. Two studies in low-income countries reported an ethnicity-based association with pneumococcal carriage [39, 41]. No other studies from low-income countries assessed ethnicity in association with carriage.

Environmental and household factors positively associated with pneumococcal carriage in low-income countries included: proxies for poverty [44, 45, 113]; rainy season [15]; dry season [105, 111]; proxy for season (study time by month) [14]; living with other children or siblings

[41–45, 105]; attendance at childcare [36, 42]; maternal pneumococcal carriage [105]; and frequency of social contacts [114].

**Factors associated with pneumococcal carriage in lower-middle-income countries.** In lower-middle-income countries, young age was associated with pneumococcal carriage [51–55, 109]. Two studies reported a positive association between male sex and pneumococcal carriage [53, 54]. Three studies reported acute respiratory illness [47, 49, 109], and one study each reported indigenous ethnicity [49] and co-colonisation with *H. influenzae* as independent predictors of pneumococcal carriage [46].

Environment and household-related risk factors for pneumococcal carriage in lower-middle-income countries included: residential location [47, 50, 54]; exposure to cigarette smoke [48, 51, 53]; proxies for poverty [52, 109]; season [50]; siblings [48, 51, 109]; attendance at childcare [52], and housing type [48].

**Factors associated with pneumococcal carriage in upper-middle-income countries.** Young age was frequently reported as positively associated with pneumococcal carriage in upper-middle-income countries [57–60, 64, 65, 99–101]. One study reported that each increasing year of age was protective against pneumococcal carriage (aOR 0.90 [95% CI 0.89–0.92]) [62]. Similarly, respiratory tract infections, variably defined as respiratory tract infection symptoms, rhinitis, infection in the month preceding survey, and previous respiratory infection, were associated with pneumococcal carriage in upper-middle-income countries [56, 57, 64, 67, 68, 100, 101].

One study reported a positive association between co-colonisation with *H. influenzae* and pneumococcal carriage in toddlers [99]. However, no association was found between co-colonisation with *H. influenzae* and pneumococcal carriage in mothers of infants and toddlers [99] or children aged under five years [59]. Similarly, one study reported co-colonisation with *Staphylococcus aureus* was negatively associated with pneumococcal carriage in infants and toddlers but was not associated with carriage in their mothers [99]. Other studies from upper-middle-income countries did not include co-colonisation of *H. influenzae* and *S. aureus* as potential risk factors in final models. Antibiotic use [63, 101] and breastfeeding [58] were protective against pneumococcal carriage. Three studies reported ethnicity-based differences in pneumococcal carriage [67, 68, 100]. One study reported physical contact with toddlers aged 12–23 months [68] and vaginal mode of infant delivery [69] as positively associated with pneumococcal carriage.

Environment and household-related risk factors for pneumococcal carriage in upper-middle-income countries included: attendance at childcare [56, 57, 63, 101] and school [58]; living with siblings or young children [56, 63, 64, 67, 69]; siblings attending childcare [60, 64]; poverty or proxies for poverty [62, 65, 67, 100]; season [99, 100], housing [59], and residential location [67].

**Factors associated with pneumococcal carriage in high-income countries.** In high-income countries, young age [66, 71–78, 80, 84, 86–88, 92–97, 103, 107, 110, 115]; ill-health, frequently involving respiratory symptoms and or otitis media [66, 73–75, 77–81, 84, 92–95, 97, 102, 104, 115]; co-colonisation with *Moraxella catarrhalis*, *H. influenzae* type b, and *S. aureus* [81, 95, 102, 104]; breastfeeding [73, 75, 97, 103]; ethnicity [83]; male sex [95, 102, 107]; and antibiotic use [66, 74–81, 84, 92, 101, 115] were found to be associated with pneumococcal carriage. One cohort study reported that at six months of age, bottle feeding was protective, compared with exclusive breastfeeding (adjusted hazard ratio 0.623 [95% CI 0.447–0.868] *P* = 0.005) [103].

Environmental and household factors associated with carriage in high-income countries included: siblings, variably defined (S5 Table) [66, 71, 73–77, 79, 81, 83, 84, 87, 88, 92–94, 97, 98, 103, 104, 107, 108, 115]; childcare attendance [66, 71–76, 78–81, 83–85, 87, 90, 92, 97, 98,

103, 104, 107, 108, 115]; proxies for poverty [75, 77, 88, 96]; residential location, defined as rural or urban, village by region of study, and community location [71, 77, 91]; exposure to cooking and cigarette smoke [86, 91, 95, 97]; and season, including autumn to winter, and summer [93, 115].

## Synthesis of results

Demographic risk factors of age [36, 41, 43, 45, 51–55, 57–60, 62, 64–66, 71–78, 80, 84, 86–88, 92–97, 99–101, 103, 107, 109–111, 113, 115]; ethnicity [39, 41, 49, 67, 68, 83, 100], symptoms of respiratory tract infection [14, 15, 38, 40, 45, 47, 49, 53, 56, 57, 64, 66–68, 73–75, 77–81, 84, 92–95, 97, 100, 101, 104, 109, 113, 115], and co-colonisation with other pathogens [15, 46, 81, 95, 99, 104] were associated with carriage in all income classifications.

Most studies that assessed the association between pneumococcal carriage and childcare attendance (either by the study participant or household member of the study participant) reported a positive association [36, 56, 57, 63, 64, 71–76, 78–81, 83–85, 87–90, 92, 93, 97, 98, 101, 103, 104, 108, 115]. However, some studies reported no association between childcare attendance and pneumococcal carriage [41, 42, 70, 85, 86, 88, 92, 93, 96, 99].

Poverty and proxies for poverty, including latrine distance from the household, number of rooms per household, crowding, malnutrition, and level of education, were found to increase the odds of carriage in all income classifications [44, 45, 52, 62, 65, 67, 75, 77, 88, 96, 100, 109, 113]. In three studies, lower socio-economic status, poverty, or proxies thereof, were not associated with carriage [63, 64, 78].

Breastfeeding and antibiotic use were protective against pneumococcal carriage in all income settings. Where assessed, most studies reported no association between breastfeeding and pneumococcal carriage [39, 41, 49, 83, 90, 97, 104, 108, 115]. Importantly, where an association between breastfeeding and pneumococcal carriage was found, it was protective against carriage [58, 73, 75, 107], except in a cohort study of Japanese infants as noted above, where bottle-feeding at six months was protective compared with breastfeeding [103].

Current and prior antibiotic use was protective against pneumococcal carriage in some low, upper-middle, and high-income country studies [14, 15, 46, 63, 66, 74–81, 84, 92, 101, 107, 109, 113, 115]. However other studies reported no association between prior antibiotic use and pneumococcal carriage [36, 37, 40, 41, 45, 47, 52, 60, 70, 78, 82, 93, 97, 104]. Further, one study reported that antibiotic use in the month preceding the survey was positively associated with carriage, but not at the time of the study [107].

In some studies, we identified some factors positively associated with carriage that were not associated with carriage in other studies. For example, the association between direct and indirect exposure to tobacco smoke and pneumococcal carriage varied. In some studies, no association between exposure to tobacco smoke and pneumococcal carriage was found [39, 41, 42, 46, 47, 60, 62, 67, 82, 83, 92, 99, 107, 108, 112]. However, where an association was found, the odds of carriage were increased by exposure to tobacco smoke [40, 48, 51, 53, 91, 97, 110, 112]. One study from Italy initially reported that maternal smoking was protective against pneumococcal carriage among children aged < 5 years [86]. However, a sub-analysis of children < 5 years from Milan only found the odds of carriage increased by greater than three-fold in association with exposure to maternal cigarette smoking (aOR [95% CI 1.53–8.73]).

In all income settings, living with young siblings or children was found to be positively associated with pneumococcal carriage in most studies (43/57, 75%) where it was assessed [41, 42, 44, 45, 47, 48, 56, 63, 66, 67, 69, 71, 73–75, 77, 79–81, 83, 84, 87, 88, 92–94, 97, 100, 103, 104, 108, 110, 112, 115]. However, findings varied between studies. Ten studies reported no association between living with young children/siblings and pneumococcal carriage [40, 46, 49, 55,

74, 78, 79, 86, 99, 108, 112]. In a cross-sectional study of Fijian infants aged 3–13 months which reported no association between living with two or more children and carriage, the lower bound of the 95% confidence interval approached a positive association [49]. Additionally, some findings within studies differed too. For example, a serial cross-sectional study, a nested longitudinal study, and a cohort study reported a positive association between siblings and pneumococcal carriage in some study years or age cohorts, but no association in other study years or age cohorts [74, 108, 112].

Where assessed, most studies found no association between participant sex and pneumococcal carriage [36–38, 40, 41, 45–47, 55, 58, 61, 62, 68, 82, 83, 88, 90, 92, 98, 108, 111, 112, 115]. However, sex-related associations with pneumococcal carriage were reported in low, middle, and high-income countries, with male sex being a risk factor in three high and two lower-middle-income countries [53, 54, 95, 102, 107] and female sex a risk factor in one low-income country [112] and one upper-middle-income country [67].

**Pneumococcal carriage rates.** S5 Table shows pneumococcal nasopharyngeal carriage rates for individual studies. Median pneumococcal carriage rates differed by income classification, ranging from 51% (IQR 19.3–70.2), 38.5% (IQR 19.3–51.6), 31.5% (IQR 19.0–51.0), 28.5% (IQR 16.75–35.4) in low-, lower-middle, upper-middle, and high-income classifications, ($\chi^2(3) = 17.742$, $P = 0.005$) (Fig 2). Outliers from high-income countries include the carriage rates reported for Indigenous Australian children aged 2–4 (82.4%) and 5–8 years (72.7%) [95], children aged 6–24 months in The Netherlands (66%) [115], and Navajo and White Mountain Apache children < 6 years in the United States of America (63.4%) [107].

Within some income classifications and countries, reported pneumococcal nasopharyngeal carriage rates were vastly different between participants of similar ages in various studies. For example, in Brazil, a pneumococcal carriage rate of 19% was reported among children aged <6 years old attending a private clinical, and 66.6% among children aged <5 years living in a slum community [57, 58]. We also noted differences in pneumococcal carriage within other countries. For example, authors reported a carriage rate of 37.2% among Gambian neonates on day 28 of life in a retrospective nested-cohort study, and 93% among neonates aged < 1 month in a cross-sectional survey of 21 villages [39, 105]. Both studies cited the 2003 WHO recommendations for detecting upper respiratory carriage of pneumococci [30]. However, in addition to differences in study design and methods, we assessed one as having good quality and the other fair (S4 Table).

## Discussion

We have identified risk and protective factors for pneumococcal carriage common to all income classifications. Factors positively associated with pneumococcal nasopharyngeal carriage in all income classifications in which they were examined included: young age [36, 41, 43, 45, 51–55, 57–60, 62, 64–66, 71–78, 80, 84, 86–88, 92–97, 99–101, 103, 107, 109–111, 113, 115]; living with young children [41, 42, 44, 45, 47, 48, 56, 63, 66, 67, 69, 71, 73–75, 77, 79–81, 83, 84, 87, 88, 92–94, 97, 100, 103, 104, 108, 110, 112, 115]; symptoms of RTI [14, 15, 38, 40, 45, 47, 49, 53, 56, 57, 64, 66–68, 73–75, 77–81, 84, 92–95, 97, 100, 101, 104, 109, 113, 115]; childcare attendance [36, 56, 57, 63, 64, 71–76, 78–81, 83–85, 87–90, 92, 93, 97, 98, 101, 103, 104, 108, 115]; ethnicity [39, 41, 49, 67, 68, 83, 100]; poverty [44, 45, 52, 62, 65, 67, 75, 77, 88, 96, 100, 109, 113]; exposure to tobacco smoke [40, 48, 51, 53, 91, 97, 110, 112]; co-colonisation with *H. influenzae*, *M. catarrhalis*, and *S. aureus* [15, 46, 81, 95, 99, 104]. Breastfeeding [58, 73, 75, 107] and antibiotic use [14, 15, 46, 63, 66, 74–81, 84, 92, 101, 107, 109, 113, 115] were found to be protective against pneumococcal carriage in most income classifications.

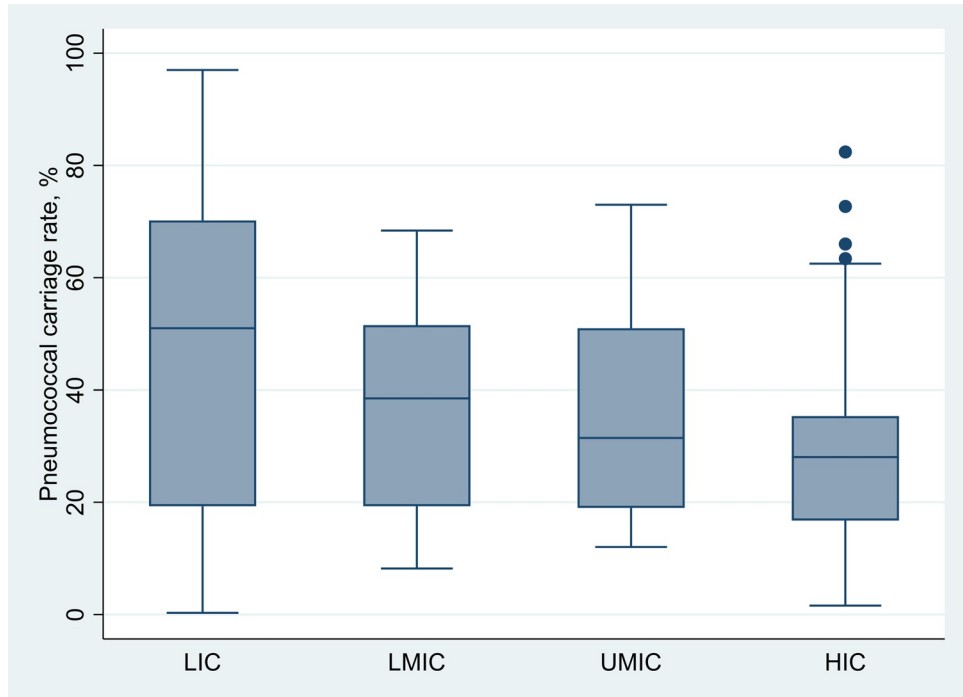

**Fig 2. Median (interquartile range) overall pneumococcal nasopharyngeal carriage rates by income classification in all ages.** Abbreviations: LIC–lower-income classification; LMIC–lower-middle-income classification; UMIC–upper-middle-income classification; HIC–high-income classification.

PCVs are the primary intervention to control pneumococcal disease [79, 117–121]. Acknowledging PCVs are the best method of pneumococcal disease prevention, introducing PCV into national immunisation programs and ensuring high uptake is important. However, pneumococcal disease persists even with the introduction of PCV. PCV reduces vaccine sero-type pneumococcal carriage in most settings. However, for unknown reasons, vaccine-sero-type carriage can persist after the introduction of PCV despite high vaccination coverage. In some LMICs, such as Malawi, vaccine-serotype carriage persisted at relatively high rates compared with high-income settings after introducing PCV13, declining from 19.9% (95% CI 15.7–25.0) to 16.7% (95% CI 13.0–21.1) among vaccinated children aged 3–5 years [19]. In Fiji, the introduction of PCV10 resulted in a relative shift to higher vaccine-serotype carriage rates in older children [68]. Despite some evidence that PCV13 reduces overall carriage prevalence in some vaccinated high-income populations, generally, PCVs do not reduce overall carriage, particularly in LMICs and indigenous populations. In addition to vaccine-serotype carriage, there are other serotypes that are not included in the vaccine and rates of non-vaccine type carriage usually increases after vaccine introduction, with some level of concomitant non-vaccine type pneumococcal disease [11, 21, 22, 122, 123]. In high pneumococcal burden and transmission settings, such as LMICs, serotype replacement disease with non-vaccine sero-types could threaten pneumococcal disease control achieved with PCVs. Therefore, it remains important to consider interventions to reduce pneumococcal carriage overall.

Vaccines and other public health interventions that modify viral respiratory pathogen infection are also likely to be important in preventing pneumococcal disease. In our review, we found that having symptoms of an acute respiratory tract infection (which are mostly viral in origin) was a risk factor for pneumococcal carriage in most studies where it was measured. Co-colonisation with respiratory viruses is associated with increased pneumococcal density,

which increases the risk of acute respiratory infections and in some settings severe pneumonia [124, 125]. In Israel, France, and South Korea, pneumococcal disease and community-acquired pneumonia declined during the public health measures used to control SARS-CoV-2 transmission [24–26]. Many countries reported declines in respiratory syncytial virus (RSV) during this period [126, 127]. In Israel, pneumonia admissions declined despite pneumococcal carriage and density remaining unchanged. However, the circulation of other co-colonising viruses which are known to increase the virulence of pneumococci declined substantially during the lockdown periods [24, 128]. Although not assessed in primary studies in this review, pneumococcal carriage is more frequent in young children during infection with RSV than with other viruses [129, 130]. Further, RSV stimulates substantial growth of pneumococci, and co-colonisation with RSV is associated with increased pneumococcal density and severity of acute respiratory tract infections [129, 131, 132]. Additionally, co-colonisation with influenza and parainfluenza have been found to increase the probability of pneumococcal acquisition [133]. This suggests that public health interventions that modify the transmission of viral respiratory pathogens are also very important in preventing pneumococcal disease, including vaccines against RSV and influenza, and other interventions that reduce viral pathogen transmission. Co-colonisation with *H. influenzae* and *M. catarrhalis* were also found to be risk factors for pneumococcal carriage in this review.

Identifying factors associated with pneumococcal carriage in certain settings may help inform other public health interventions that may be needed. Some risk factors are not modifiable, such as age, living with young children (however this is most likely due to increased viral transmission in this age group), and ethnicity. However, the risk of pneumococcal carriage, transmission, and disease may be reduced by public health programs and policies that target particular age groups [134, 135], to reduce transmission, such as, such as increased access to improved sanitation and hygiene [136, 137], or that are tailored to address socio-economic differences and social determinants of health which promote tranmission [138]. Reducing environmental risk factors for pneumococcal carriage and viral transmission includes improving breastfeeding, reducing malnutrition, preventing overcrowding, enhancing respiratory etiquette, and reducing smoke exposure. Public health programs that promote birth spacing (which may reduce the number of young siblings living in the same household), breastfeeding, interventions to reduce poverty, and which ensure high coverage of infant vaccination, may reduce the risk of pneumococcal disease [24–26]. Many of these modifiable factors are included in the WHO integrated Global Action Plan for the Prevention and Control of Pneumonia and Diarrhoea [139]. Having programs to address these factors would also help prevent other infectious diseases that are a common cause of child morbidity and mortality in LMICs.

There was considerable heterogeneity in the factors collected and the definitions of the factors included in studies. Consequently, the absence of reported associations between potential risk factors and pneumococcal carriage may be because factors were not measured or included in models. For example, studies in high-income countries frequently measured childcare attendance or similar. However, this was measured in three low- [36, 41, 42] and one lower-middle-income country [52]. The heterogeneity of collected data hindered the comparison of rates of risk factors across income classifications and countries. Most studies were conducted in high-income countries, yet the burden of pneumococcal disease is highest in low- and lower-middle-income countries [4]. The relatively limited risk factors reported from low- and lower-middle-income classifications reflect the limited number and range of studies investigating risk factors for pneumococcal carriage in such settings, rather than a lack of risk factors.

Pneumococcal carriage rates varied by income classification and within countries. The highest pneumococcal carriage rate was in low-income classifications. Higher carriage rates may be due to higher rates of risk factors in low-income classifications compared with high-

income classifications. This systematic review has brought together diverse studies from around the globe. For example, diversity is evident in the quality, inclusion criteria, study duration, and methods of pneumococcal detection and risk factor assessment. A meta-regression to understand drivers of variation in carriage across studies would have been possible had we comparable population-level information on studies. However, few studies measured the same exposure variables or measured them with similar definitions, and population-level risk factors were not documented, preventing comparison of risk factor rates by income classifications, and limiting conclusions that could be drawn regarding differences in overall carriage rates across populations.

Nonetheless, there was some evidence of higher rates of risk factors associated with pneumococcal carriage, such as living with siblings or young children, in low-income classifications compared with high-income classifications. Further, some factors associated with pneumococcal carriage, such as exposure to tobacco smoke, are known to have higher rates in LMICs compared with HICs. In 2021, the WHO reported 1.3 billion tobacco users, 80% of whom live in LMICs [140]. A complex relationship between multiple intertwined factors may explain the higher rates of pneumococcal carriage in low-income classifications. Pneumococcal carriage rates also varied between studies within the same countries, perhaps due to differences in sampled populations, the prevalence of potential risk factors for pneumococcal carriage [57, 58]; age groups sampled [14, 44, 51, 58, 86, 87, 100] study designs and microbiological methods [30, 31, 72, 107]; and study quality [39, 105]. Indigenous ethnicity was identified as a risk factor for pneumococcal carriage [67, 68]. Further, notably high carriage rates were identified in Indigenous Australian and Navajo and White Mountain Apache children in the United States of America [95, 107]. Social determinants of health likely affect differential pneumococcal carriage (and disease) burden within countries, however comprehensive analysis of factors predisposing towards differences in pneumococcal carriage between indigenous and non-indigenous populations living in the same settings remain largely unqualified and unquantified.

## Limitations

This review has some limitations. Although articles from low- and lower-middle-income countries were included in this review, most primary studies were conducted in high-income countries. Low- and lower-middle-income countries were proportionally underrepresented, limiting the potential representativeness of studies for these income settings. Further, most studies used convenience sampling. For these reasons, the studies for which pneumococcal nasopharyngeal carriage rates were available may not be representative of regional, country, or within-country populations. Therefore, we caution against using the reported rates by income classification as population or sub-population rates. The methods used to detect pneumococcal nasopharyngeal carriage differed, and most studies did not report using the WHO guidelines [31]. Consequently, differences in swab materials transport, storage, and laboratory methods may impact the measurement of pneumococcal carriage in addition to reported risk factors. Further, our review was not a systematic review of rates of pneumococcal carriage. Instead, we opportunistically used data reported in studies reporting risk factors associated with carriage in varying age groups. This may mean that the summary of reported pneumococcal carriage rates by income classification may not be representative. Additionally, variables were not collected uniformly in each study, and those selected *a priori* for inclusion in models to assess factors associated with pneumococcal carriage differed by study. Due to the heterogeneity of studies, assessment of publication bias by quantitative methods was not possible.

## Conclusion

PCVs are the primary intervention against pneumococcal disease. Interventions to reduce viral transmission, including RSV vaccine, are also likely to play an important role in preventing pneumococcal disease. This global systematic review fills a knowledge gap regarding risk factors for pneumococcal nasopharyngeal carriage. This review identifies factors to consider for inclusion in future carriage studies, including participant age, ethnicity, symptoms of respiratory tract infection at the time of the survey, childcare attendance and living with other young children, poverty or proxies for poverty, exposure to cooking, and cigarette smoke, season, co-colonisation with other pathogens, breastfeeding, and antibiotic use. Public health programs to address these factors would also help prevent other common infectious diseases in children, especially those living in disadvantage, and help prevent morbidity and mortality. Standard definitions and measurements of variables included in future studies would be preferable to aid comparison. Similarly, future studies should comply with the WHO methods for detecting pneumococci in the upper respiratory tract [31]. Finally, this review summarises hypothesis-generating studies and may inform future studies designed to ascertain causal relationships between factors identified as associated with pneumococcal carriage.

## Supporting information

**S1 Checklist. PRISMA checklist.**
(DOCX)

**S1 Text. Search strategies.**
(DOCX)

**S1 Table. Summary of studies reporting risk factors for pneumococcal carriage, stratified by World Bank income status, WHO region, and country.**
(DOCX)

**S2 Table. Exposure variables collected among participants included in studies reporting risk factors for pneumococcal carriage, stratified by World Bank income status, WHO region, and country.**
(DOCX)

**S3 Table. Details of laboratory methods in studies reporting risk factors for nasopharyngeal pneumococcal carriage, stratified by World Bank income status, WHO region, and country.**
(DOCX)

**S4 Table. Quality assessment of included studies.**
(DOCX)

**S5 Table. Factors assessed for association with pneumococcal nasopharyngeal carriage, stratified by World Bank income status, WHO region, and country.**
(DOCX)

**S1 Data. Study quality assessment data.**
(DTA)

**S2 Data. Pneumococcal carriage data.**
(DTA)

## Author Contributions

**Conceptualization:** Eleanor Frances Georgina Neal, Fiona Mary Russell.

**Data curation:** Eleanor Frances Georgina Neal, Jocelyn Chan.

**Formal analysis:** Eleanor Frances Georgina Neal.

**Investigation:** Eleanor Frances Georgina Neal.

**Methodology:** Eleanor Frances Georgina Neal.

**Project administration:** Eleanor Frances Georgina Neal.

**Supervision:** Cattram Duong Nguyen, Fiona Mary Russell.

**Visualization:** Eleanor Frances Georgina Neal.

**Writing – original draft:** Eleanor Frances Georgina Neal.

**Writing – review & editing:** Eleanor Frances Georgina Neal, Jocelyn Chan, Cattram Duong Nguyen, Fiona Mary Russell.

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
