## [Decision Letter · Decision Letter 0]

31 Jan 2022

PGPH-D-21-00990

Factors associated with pneumococcal nasopharyngeal carriage: a systematic review

Dear Eleanor F G Neal ,

Thank you for submitting your manuscript to PLOS Global Public Health. After careful consideration, we feel that it has merit but does not fully meet PLOS Global Public Health’s publication criteria as it currently stands. Therefore, we invite you to submit a revised version of the manuscript that addresses the points raised during the review process.

Please respond in detail to the reviewer comments. Please submit your revised manuscript by 30 days. If you will need more time than this to complete your revisions, please reply to this message or contact the journal office at globalpubhealth@plos.org. Please include the following items when submitting your revised manuscript:

Reviewers comments : 

Reviewer 1: Thank you for inviting me to review this paper. A well written and comprehensive review. However, I suggest minor revision for this review. Find below my comments:

1. Line 48: Write out the full meaning of HIV

2. Line 62: Any reason why facility-based studies were not included?

3. Line 73: Any reason for not including grey literatures?

4. Line 88: Full meaning of WHO.

5. Line 97: Give the basis for the use Kruskal-Wallis method for this analysis.

6. Line 120: In the discussion section can you give likely reasons for having just only one RCT in this study.

7. Line 147: Any statement on the poor-quality studies? Were they included in the final analysis? Reasons for inclusion or non-inclusion.

8. Discussion – Need to make specific public health recommendations based on the findings of this review.

9. Line 384: There is need to include the “References” heading.

10. Figure 1 – This is blur. Include a clearer picture.

Reviewer 2: This systematic review describes some factors associated with pneumococcal carriage in children by classification of the settings on the income of country. It also describes carriage rates in children across those country level income categories. The manuscript is well structured and clearly written. The methods are acceptable and clearly described in adequate detail. I have the following comments to make.

A systematic review to answer this question need to rely on what RFs are assessed in the primary studies and that will not obviously be similar across studies. This needs to be properly highlighted and needs to be addressed in the methods with mention of any attempt to select homogenous studies in this regard somehow (noting studies in LMIC are few).

How representative are these studies for each income category setting in particular in the low income settings. Even within these countries there could be variation between regions and population groups.

I get the sense that authors somewhat downplay the impact of PCV use in changing the pneumococcal carriage levels. It is true that in some settings NVTs have replaced VTs in carriage with not a substantial decline overall. However, there is overwhelming evidence that in all settings there is large reductions in disease particularly that of severe end of the spectrum (IPD). The classification of these study settings needs to be considered in terms of PCV use, schedule and duration of program.

Authors in the discussion highlight the importance on non-vaccine interventions to reduce carriage rates. I think these are all secondary. Besides are there sufficient evidence to suggest a significant reduction in carriage that causes most disease as a result of these possible ‘other’ interventions (noting also that most risk factors identified are not modifiable). This needs to be addressed in the discussion.

The reason for excluding studies assessing RFs associated with VT and non-VT type carriage is not clear.

Need some mention of carriage in First Nations/Indigenous population in high income countries some of whom might have living conditions somewhat similar to LMIC settings.

There are some conclusions drawn regarding the differences in carriage rates across income settings. The key question addressed in the review is risk factors associated with carriage primarily at an individual level. Drawing conclusions on overall carriage rates would need to consider the prevalence of risk factors in respective population. This needs to be addressed.

We look forward to receiving your revised manuscript.

Kind regards,

Nusrat Homaira

Academic Editor

---

## [Editor Report · Decision Letter 1]

14 Mar 2022

Factors associated with pneumococcal nasopharyngeal carriage: a systematic review

PGPH-D-21-00990R1

Dear  Eleanor F G Neal

We are pleased to inform you that your manuscript 'Factors associated with pneumococcal nasopharyngeal carriage: a systematic review' has been provisionally accepted for publication in PLOS Global Public Health.

Best regards,

Nusrat Homaira

Academic Editor